# Discovering Highly Influential Shortcut Reasoning:
# An Automated Template-Free Approach

**Daichi Haraguchi[1], Kiyoaki Shirai[1], Naoya Inoue[1,2], Natthawut Kertkeidkachorn[1]**
[1]Japan Advanced Institute of Science and Technology
[2]RIKEN
{s2110137,kshirai,naoya-i,natt}@jaist.ac.jp

## Abstract

Shortcut reasoning is an irrational process of inference, which degrades the robustness of an NLP model. While a number of previous work has tackled the identification of shortcut reasoning, there are still two major limitations: (i) a method for quantifying the severity of the discovered shortcut reasoning is not provided; (ii) certain types of shortcut reasoning may be missed. To address these issues, we propose a novel method for identifying shortcut reasoning. The proposed method quantifies the severity of the shortcut reasoning by leveraging out-of-distribution data and does not make any assumptions about the type of tokens triggering the shortcut reasoning. Our experiments on Natural Language Inference and Sentiment Analysis demonstrate that our framework successfully discovers known and unknown shortcut reasoning in the previous work.[1]

## 1 Introduction

While Transformer-based large language models have remarkably improved various NLP tasks, the issue of *shortcut reasoning* has been identified as a severe problem (Schlegel et al., 2020; Wang et al., 2022b; Ho et al., 2022). Shortcut reasoning usually refers to the irrational inference of a model, which is derived from spurious correlations in the training data (Gururangan et al., 2018; Poliak et al., 2018; McCoy et al., 2019). For example, sentiment analysis models could learn to classify any sentences containing the word *Spielberg* into POSITIVE, given a training dataset with many positive movie reviews containing *Spielberg* (e.g., *Spielberg is a great director!*).

Shortcut reasoning makes models brittle against Out of Distribution (OOD) data (i.e., data from a different distribution from the training data) compared to Independent and Identically Distributed

(IID) data (i.e., data from an identical distribution as the training data) (Geirhos et al., 2020). In the aforementioned example, the reasoning for movie reviews would not be valid for OOD data (e.g., news articles) because the sentiment of news articles containing *Spielberg* could be arbitrary.

Although many studies have explored the detection of spurious correlations or shortcut reasoning (Ribeiro et al., 2020; Pezeshkpour et al., 2022; Han et al., 2020), several challenges persist. Wang et al. (2022a) propose a state-of-the-art method for discovering shortcut reasoning, which implements an automated framework to discover shortcuts without predefining shortcut templates. Still, their approach suffers from two major limitations.

Firstly, their framework lacks a method for quantifying the severity of the discovered shortcut reasoning on OOD data. Even if the shortcuts are identified, we do not have to necessarily be concerned about them as long as they have little negative impact on the model's robustness. Secondly, their approach assumes that *genuine* tokens, useful tokens for predicting labels across different datasets (e.g., "good", "bad"), do not lead to shortcut reasoning. While this assumption seems reasonable, Joshi et al. (2022) argue that such tokens are still prevalent among spurious correlations. This is because such tokens are indeed necessary to predict the label, but these tokens alone may not provide sufficient information to accurately predict labels. For example, a genuine token "good" in a sentence *This movie is not good* can be spurious, since "good" is a necessary but insufficient token for determining its sentiment label. Therefore, genuine tokens can not be ignored when identifying shortcut reasoning.

To address these problems, we propose a new method for discovering shortcut reasoning. Our contributions can be summarized as follows:

- We present an automated method for identifying shortcut reasoning.

---

[1]Our code is available at https://github.com/homoscribens/shortcut_reasoning.git

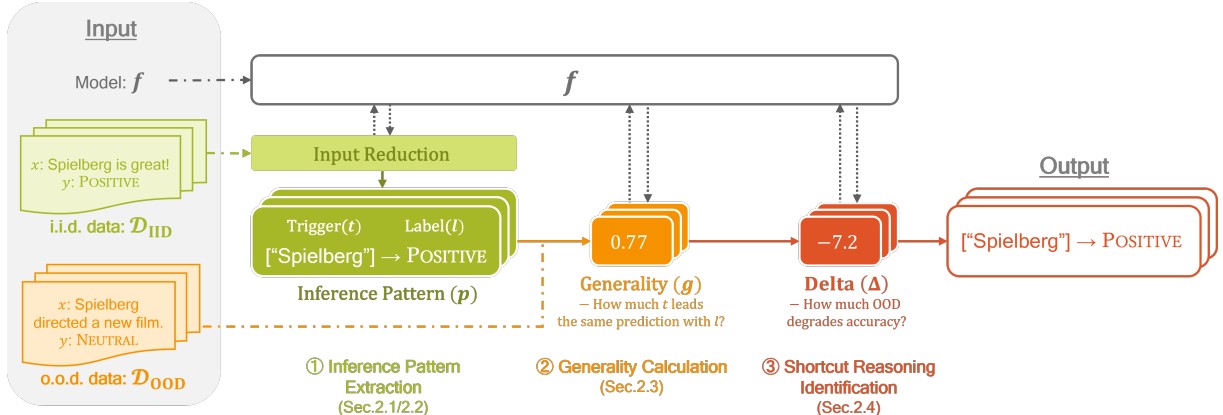

Figure 1: Our method to discover shortcut reasoning with 3 steps.

- By applying the less subjective definition of shortcut reasoning (Geirhos et al., 2020), our method does not require laborious human evaluation for detected shortcuts.

- Our method quantifies the severity of shortcut reasoning by leveraging OOD data and does not make any assumptions about the type of tokens triggering the shortcut reasoning.

- We demonstrate that our method successfully discovers previously unknown shortcut reasoning as well as ones reported in previous research.

## 2 Discovering Shortcut Reasoning

Fig. 1 shows the overall procedure of the proposed method. Given (i) a target model $f$, (ii) IID data $\mathcal{D}_{\text{IID}}$, and (iii) OOD data $\mathcal{D}_{\text{OOD}}$ as inputs, shortcut reasoning is extracted as an output. The procedure consists of the following three steps.

**Step 1** extracts *inference patterns*, an abstract representation that characterizes the inferential process of a given model (§2.1). To extract inference patterns, we use *input reduction*, an algorithm that automatically derives the inference patterns (§2.2). **Step 2** estimates the *generality* of the extracted inference patterns. Generality is a measure of the strength of an inference pattern, indicating its degree of regularity (§2.3). **Step 3** identifies shortcut reasoning. We automatically determine whether an inference pattern exhibits shortcut reasoning without human intervention, comparing the effectiveness of inference patterns in $\mathcal{D}_{\text{IID}}$ with that in $\mathcal{D}_{\text{OOD}}$ and leveraging the estimated generality as a proxy for the severity of the identified shortcut reasoning (§2.4).

### 2.1 Inference Pattern

An inference pattern is a crucial pattern that activates a label during a model's inference process. Given a target model $f$, we formally define an inference pattern $p$ as follows:

$$p \overset{\text{def}}{=} t \overset{f}{\rightarrow} l, \tag{1}$$

where $t$ denotes a trigger that induces a certain label, and $l$ is the induced label.

Pezeshkpour et al. (2022) classified spurious correlations into two types: (i) granular features, namely discrete units such as an individual token "Spielberg", and (ii) abstract features, namely high-level patterns such as lexical overlap.

This paper focuses on granular features and leaves the detection of shortcut reasoning with abstract features for future work. We thus adopt the following definition as an inference pattern:

$$p \overset{\text{def}}{=} \mathbf{w} \overset{f}{\rightarrow} l, \tag{2}$$

where $\mathbf{w}$ is a sequence of tokens $[w_1, w_2, \cdots, w_n]$.

Although we limit ourselves to granular features, this definition still enables us to detect shortcut reasoning with a variety of forms, such as combinations of tokens as well as a single token. For example, a sentiment analysis model $f$ may have inference patterns such as ["not", "bad"] → NEUTRAL or ["Spielberg"] → POSITIVE (possibly shortcut reasoning).

### 2.2 Extracting Inference Patterns

Given a target model $f$ and an IID dataset $\mathcal{D}_{\text{IID}} = \{(x_i, y_i)\}_{i=1}^{N}$, we extract a set $C$ of inference patterns by applying input reduction (IR) to each input $x_i$.

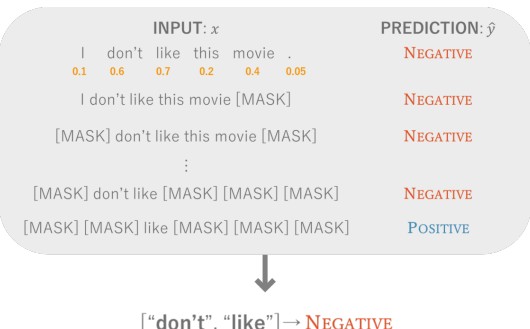

Figure 2: An example of inference pattern extracted by Input Reduction.

IR gradually reduces the number of tokens in $x_i$ by masking each token one by one, incrementally increasing the number of masked tokens after each step. In each step, IR feeds the masked $x_i$ into $f$, and obtains a predicted label $\hat{y}_i$. IR stops when the predicted label $\hat{y}_i$ flips. As the final step, IR extracts a sequence $\mathbf{w}_i$ of unmasked tokens in $x_i$ and $\hat{y}_i$ that is the last predicted label before the prediction flips as an inference pattern, namely $\mathbf{w}_i \xrightarrow{f} \hat{y}_i$.

To prioritize which tokens should be masked, we employ Integrated Gradient (Sundararajan et al., 2017), which computes the importance of each token for prediction. IR sorts the tokens in $x_i$ according to their IG score and then incrementally applies masks to the tokens in ascending order of their rank in the sorted sequence. For each mask applied, IR leaves the corresponding token masked and proceeds to the next token in the sequence.

Fig. 2 shows an example of the extracting process by IR. The tokens are replaced with [MASK] in the ascending order of the IG scores (values in orange in the figure). When the predicted label is flipped from NEGATIVE to POSITIVE, the remained tokens at the previous step and NEGATIVE label are extracted as the inference pattern ["don't", "like"] → NEGATIVE. See Appendix A for the pseudocode of IR.

The IR-based extraction algorithm ensures that the trigger $\mathbf{w}_i$ of the extracted inference patterns is concise and not redundant.

## 2.3 Calculating Generality

In order to verify the validity of an inference pattern in $C$ as a universal pattern, we assess its generality on $\mathcal{D}_{\mathrm{OOD}}$. This measurement determines the degree to which the pattern exhibits regularity in the OOD dataset.

To estimate the generality of inference pattern

$p_i = \mathbf{w}_i \to l_i \in C$, we collect a set $E_{\mathrm{OOD}}(\mathbf{w}_i)$ of examples from $\mathcal{D}_{\mathrm{OOD}}$ such that the input contains $\mathbf{w}_i$. For example, given $p_i = [\text{"Spielberg"}] \to$ POSITIVE, $E_{\mathrm{OOD}}(\mathbf{w}_i)$ may contain sentences such as *I grew up with Steven Spielberg's films. His films are always great!!* and *Spielberg is overrated.* We then estimate the generality $g$ of the inference pattern $p_i$ as follows:

$$g(p_i) \overset{\text{def}}{=} \frac{\sum_{x' \in E_{\mathrm{OOD}}(\mathbf{w}_i)} \mathbb{1}\left[f(x') = l_i\right]}{|E_{\mathrm{OOD}}(\mathbf{w}_i)|} \times 100. \quad (3)$$

Intuitively, $g(p_i)$ explains how much the inference pattern is dominant on the OOD dataset.

## 2.4 Identifying Shortcut Reasoning

In this section, we define shortcut reasoning and describe the method for its detection. According to Geirhos et al. (2020), shortcut reasoning satisfies both of the following conditions: (i) performs well on $\mathcal{D}_{\mathrm{IID}}$, and (ii) underperforms on $\mathcal{D}_{\mathrm{OOD}}$.

We apply these conditions to inference patterns. Given $p_i = \mathbf{w}_i \to l_i$ extracted from $\mathcal{D}_{\mathrm{IID}}$ by IR, the condition (i) is satisfied when $p_i$ works well on $\mathcal{D}_{\mathrm{IID}}$. In other words, when the model performs well on IID examples that contain $\mathbf{w}_i$ (i.e., $E_{\mathrm{IID}}(\mathbf{w}_i)$). Thus, we evaluate the performance of each inference pattern using $E_{\mathrm{IID}}(\mathbf{w}_i)$. Specifically, we define a new metric iid_acc$_i$, which computes

$$\frac{\sum_{x \in E_{\mathrm{IID}}(\mathbf{w}_i)} \mathbb{1}\left[f(x) = l_i \wedge l_i = y_i\right]}{\sum_{x \in E_{\mathrm{IID}}(\mathbf{w}_i)} \mathbb{1}\left[f(x) = l_i\right]} \times 100. \quad (4)$$

This metric counts the right prediction for inputs that contain trigger ($\mathbf{w}_i$).

The condition (ii) is satisfied when $p_i$ does not deliver accurate results on $\mathcal{D}_{\mathrm{OOD}}$, i.e., when the model operates poorly on OOD inputs that contain $\mathbf{w}_i$ (i.e., $E_{\mathrm{OOD}}(\mathbf{w}_i)$). As a metric to evaluate how much the $p_i$ underperforms over $E(\mathbf{w}_i)$, we define $\Delta$ as follows:

$$\Delta_i \overset{\text{def}}{=} \mathrm{F1}(E_{\mathrm{OOD}}(\mathbf{w}_i), f) - \mathrm{F1}(\mathcal{D}_{\mathrm{OOD}}, f). \quad (5)$$

This metric compares the F1 score on $E_{\mathrm{OOD}}(\mathbf{w}_i)$ to that on $\mathcal{D}_{\mathrm{OOD}}$, employed as a baseline for comparison.

To sum up, shortcut reasoning is defined as $\tilde{p}_i = \mathbf{w}_i \to l_i$ such that $g(p_i)$ is sufficiently large, iid_acc is large enough and $\Delta_i$ is small enough. The set $\tilde{P}$ of shortcut reasoning is defined as

$$\{p_i \in C \mid g(p_i) > \lambda_1, \text{iid\_acc}_i > \lambda_2, \Delta_i < \lambda_3\}. \quad (6)$$

$\lambda_1$, $\lambda_2$, and $\lambda_3$ are the pre-defined thresholds. Note that $\lambda_2$ and $\lambda_3$ have to be an above-chance score and less than 0 at least, respectively. This definition enables us to automatically identify shortcut reasoning that has a substantial impact on OOD, unlike previous studies (Pezeshkpour et al., 2022; Wang et al., 2022a).

## 3 Experiments

### 3.1 Setup

One straightforward approach for assessing our method is to annotate NLP models with their ground-truth shortcut reasoning. However, recent NLP models are known to be hard to interpret, which makes it difficult to create such a reference dataset. We thus resort to existing datasets for Natural Language Inference (NLI) and Sentiment Analysis (SA) that have been shown to contain spurious features and check if our inference patterns can reveal such features (and unknown ones).

**Datasets** For NLI, we adopt MNLI (Williams et al., 2018) as $\mathcal{D}_{\text{IID}}$ and ANLI (Nie et al., 2020) as $\mathcal{D}_{\text{OOD}}$. ANLI is an NLI dataset based on MNLI, but adversarially redesigned, which makes it harder to answer. For SA, we apply Sentiment subset in Tweeteval (Barbieri et al., 2020) as $\mathcal{D}_{\text{IID}}$ and MARC (Multilingual Amazon Reviews Corpus) (Keung et al., 2020) as $\mathcal{D}_{\text{OOD}}$. For all OOD datasets, we use training split of each. See Appendix B.1 for the dataset details. Note that all the datasets are trinary classifications, so the chance accuracy is 0.33.

**Models** We apply our method to RoBERTa (Liu et al., 2019) fine-tuned with the $\mathcal{D}_{\text{IID}}$ mentioned above, available at Hugging Face Hub. See Appendix B.2 for the details.

**Configuration** The inference patterns of a model are obtained by learning training data. Thus, aside from test (or validation) sets, we extract $C$ from the training set of $\mathcal{D}_{\text{IID}}$, expecting to better simulate the model's reasoning process. In addition, we randomly choose 1,000 examples as input to IR considering its runtime. We set hyperparameters $\lambda_1 = 50$, $\lambda_2 = 70$ and $\lambda_3 = -0.05$. To reliably obtain $g(p_i)$, we filter out $p_i$ with $|E_{\text{OOD}}(\mathbf{w}_i)| < 100$ from $C$.

### 3.2 Results

We show samples of the results in Table 1. We selected representative $\tilde{p}$, which have large $g$, iid_acc, and $|\Delta|$. The column train/test denotes whether shortcut reasoning is discovered in the training or the test split of IID dataset (i.e., input for IR).

**NLI** For OOD data, the model performed 77.8 of $\text{F1}(\mathcal{D}_{\text{OOD}}, f)$. "/s" denotes a separation token between premise and hypothesis. For OOD data, the model performed 77.8 of $\text{F1}(\mathcal{D}_{\text{OOD}}, f)$. We observed that most of $t$ identified as shortcut reasoning belonged to the hypothesis, while only a small proportion was present in the premises. This observation suggests that the model heavily relies on the hypothesis to predict labels, corroborating the findings of Poliak et al. (2018). Furthermore, we found that negation representation in $t$, such as "not" or "never", often led the model to predict CONTRADICTION. This phenomenon manifests itself even when the gold label indicates otherwise, as indicated by the values of $\Delta$ at the first and fourth $\tilde{p}$ in the Table 1. This finding aligns with the results reported by Gururangan et al. (2018). With this observed consistency with previous work, our method seems to be effective at accurately identifying shortcuts.

**SA** The model showed 60.3 points at $\text{F1}(\mathcal{D}_{\text{OOD}}, f)$. We found that sentiment words, such as "worst" or "Excellent", emerged in almost all $t$ that were classified as shortcut reasoning. Further analysis showed that reviews with neutral labels in MARC frequently contained both positive and negative sentiments (e.g., *I hate the wrapping, but it works pretty well.*). Considering a sufficient number of $p$ with small $\Delta$ are annotated with neutral label for the original input, we estimate that the model relies on one among multiple sentiments in the input and ignores the rest. Therefore, it is possible to say that these inference patterns are shortcuts, whose $t$ are necessary but insufficient.

**Train/Test** No significant difference was observed between the train and test experiments. Although the extracted shortcut reasoning from the experiments differed, they were essentially similar in terms of their characteristics (such as negations in NLI or sentiment words in SA).

**Unknown Shortcut Reasoning** In the NLI experiment, it is interesting to note that we revealed several previously unknown shortcut reasoning, such as ["soon"] $\rightarrow$ NEUTRAL and ["is", "always"] $\rightarrow$ NEUTRAL. Both have sufficiently small $\Delta$, and large iid_acc and $g$ to be considered as $\tilde{p}$.

| | $\tilde{p}$ | $g$ | iid_acc | $\Delta$ | $|E_{\mathrm{IID}}(\mathbf{w}_i)|$ | train/test |
|---|---|---|---|---|---|---|
| **NLI** | ["/s", "never" ]→ CONTRADICTION | 80.3 | 99.3 | -9.0 | 1515 | ✓/✓ |
| | ["/s", "soon" ]→ NEUTRAL | 64.1 | 100.0 | -6.5 | 142 | ✓/ ✓ |
| | ["/s", "is", "always" ]→ NEUTRAL | 60.8 | 96.6 | -5.2 | 102 | ✓/ - |
| | ["/s", "not" ]→ CONTRADICTION | 55.0 | 94.5 | -55.6 | 8708 | ✓/✓ |
| **SA** | ["worst" ]→ NEGATIVE | 97.5 | 81.7 | -25.4 | 158 | ✓/ - |
| | ["Excellent" ]→ POSITIVE | 96.2 | 100.0 | -7.2 | 184 | - /✓ |
| | ["Perfect" ]→ POSITIVE | 96.0 | 88.6 | -12.9 | 324 | - /✓ |
| | ["poor" ]→ NEGATIVE | 87.1 | 76.0 | -12.9 | 458 | ✓/ - |

Table 1: Samples of identified shortcut reasoning on NLI (above) and SA (bottom). We show several interesting results and $\tilde{p}$ that holds large $g$ and iid_acc and small $\Delta$.

## 4 Related work

Numerous studies have tackled the problem of detecting spurious correlations or shortcut reasoning (Ribeiro et al., 2020; Han et al., 2020). One major limitation of earlier studies is that they predefine a specific format or structure for the shortcuts (e.g., a single token or a predefined set of tokens). This can hinder the discovery of new and unexplored shortcuts, which may be manifested in diverse forms.

Recently, Pezeshkpour et al. (2022) address this problem by combining multiple interpretability techniques, such as influence function (Koh and Liang, 2017) and feature attribution methods (e.g., Integrated Gradient). However, they rely on human assessment to identify shortcut reasoning, which can result in misjudgment between rational and irrational reasoning. Besides, human evaluation is laborious and time-consuming.

Wang et al. (2022a) solve this issue by automatically identifying genuine tokens, important tokens that appear across different datasets, and spurious tokens, important tokens that appear only in an in-domain dataset. Still, as discussed in §1, there is a major limitation with their approach in that they fail to consider the influence of the identified shortcut reasoning on OOD data. Our work attempts to address this issue by estimating the generality of inference patterns. Besides, our definition of shortcut reasoning aligns well with more practical scenarios. While they rely on a subjective definition of shortcut reasoning (i.e., whether reasoning is irrational from humans' point of view), our work targets shortcut reasoning that performs well on IID but underperforms on OOD (Geirhos et al., 2020), namely the one that clearly hurts the robustness of NLP models by definition.

## 5 Conclusion

We introduced a method to automatically discover shortcut reasoning. With minimal predefinition, our method successfully identified known and previously unknown examples of shortcut reasoning. For future research, we plan to adapt our method to large language models and other tasks such as machine reading comprehension. Overall, we hope that our study provides a promising approach towards understanding the behavior of deep learning models and improving their trustworthiness.

## 6 Limitations

Firstly, we have yet to develop an evaluation process to validate the discovered shortcut reasoning. Even though we indicate the metrics or measurement of shortcut reasoning, knowing the actual reasoning process is impossible if we use black box models. Unfortunately, this problem would require significant effort to be solved.

Secondly, as our method is not compatible with abstract inference patterns, it cannot cover all kinds of shortcut reasoning other than the granular one.

Thirdly, preparing two datasets, i.e., IID and OOD, is challenging for low-resource languages or some tasks. This problem limits the further studies or application of this method. Fortunately, now that we can access large language models that have surprising linguistic capabilities and are well-aligned with the user's instruction. The generated examples by LLMs have a certain distribution which can be treated as OOD for target models, or we can prompt them to generate examples with specific distribution.

The fourth limitation is about IR. If the prediction for the masked input does not flip during the

reduction, then we alternatively output the last token left in the input. Therefore, in some cases, we cannot guarantee that the extracted inference pattern is genuine.

Finally, input reduction can be utilized only when a MASK token is available on the input model.

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

## A  Details of Input Reduction

---
**Algorithm 1** Pseudo-code of Input reduction

---
1: **function** INPUT_REDUCTION_IG($\mathcal{D}$)
2:     **for all** $(x, y) \in \mathcal{D}$ **do**
3:         $\hat{y} \leftarrow f(x)$ ; $x' \leftarrow x$ ; $\hat{y}' \leftarrow \hat{y}$
4:         **while** $\hat{y} = \hat{y}'$ **do**
5:             $x'_{prev} \leftarrow x'$ ; $\hat{y}'_{prev} \leftarrow f(x'_{prev})$
6:             $x' \leftarrow$ IG_mask($x'$)
7:             $\hat{y}' \leftarrow f(x')$
8:             **if** all tokens in $x'$ are mask **then**
9:                 break
10:             **end if**
11:         **end while**
12:         $C \leftarrow C \cup \{p = (x'_{prev}, \hat{y}'_{prev})\}$
13:     **end for**
14:     **return** $C$
15: **end function**

---

## B  Experimental setup

### B.1  Dataset detail

| Dataset | train | validation | test |
|---|---|---|---|
| **NLI** | | | |
| MNLI (matched) | 392,702 | 9,815 | 9,796 |
| ANLI (round3) | 100,459 | 1,200 | 1,200 |
| | | | |
| **SA** | | | |
| Tweeteval (sentiment) | 45,615 | 2,000 | 12,284 |
| MARC (en) | 200,000 | 5,000 | 5,000 |

### B.2  Models detail

| Task | Model |
|---|---|
| **NLI** | roberta-large-mnli |
| **SA** | cardiffnlp/twitter-roberta-base-sentiment |