# OpenReview forum: "Discovering Highly Influential Shortcut Reasoning: An Automated Template-Free Approach"
_EMNLP/2023/Conference — EMNLP 2023 Findings_

### Official Review · Reviewer_xMKL · 2023-08-03

**Soundness:** 3

**Excitement:**

3: Ambivalent: It has merits (e.g., it reports state-of-the-art results, the idea is nice), but there are key weaknesses (e.g., it describes incremental work), and it can significantly benefit from another round of revision. However, I won't object to accepting it if my co-reviewers champion it.

**Paper Topic And Main Contributions:**

This paper proposes a method to automatically discover granular-feature shortcut reasoning in NLP models and quantitatively evaluate how much the shortcut reasoning affects the model's robustness. Compared to previous works, this method does not rely on any certain heuristic assumptions about the trigger token. Experiment observations are consistent with previous works. The authors also find some shortcuts that are previously unknown.

**Questions For The Authors:**

### Question about method details

1. In line 127, $\textbf{w}$ is a sequence of words, while in line 146, $\textbf{w}_i$ is a sequence of unmasked tokens. As for my understanding, a 'word' is the basic element of language that carries an objective or practical meaning and a 'token' is the result of any tokenizer (which could be subword tokens, e.g. roberta, the model used in the experiment, uses a byte-level BPE as a tokenizer). Which one is used in the paper? The terminology is not consistent.
2. Line 155, Section 2.2. Whether the most important token is masked first, or the least one? According to the pseudocode, IR-based extraction algorithm recognize the pattern as all the tokens left in the sentence before the label flips. So it should filter out the least important tokens, right?
3. Appendix A. According to the algorithm, if the label does not change all the time, then the last token will still be recognized as a pattern. The limitation also mentions this. Why? I think it is more natural to drop this sample and not produce any pattern.
4. Appendix A. According to the algorithm, what will happen if $\hat y \ne \hat y'$ even if no token is masked? i.e. wrong prediction in the IID dataset. Based on line 202, I guess these samples won't be taken into consideration. But it would be better if stated clearly.

### Question about soundness

1. Line 189, Section 2.4. When $l_i$ meets its gold label $y_i$, the model performs well on $D_{IID}$. Why? I think only one example does not support the claim.
2. Relatedly, line 255, Section 3.2. Negation expressions are known to be difficult for NLP models. For IID dataset, the performance on samples with negation expressions will also be much lower than other samples[1]. Also, according to table 1, g is not very high, which may indicate that the model is not very confident at these samples rather than taking it as a shortcut.

[1] Naik, A., Ravichander, A., Sadeh, N., Rose, C., & Neubig, G. (2018). Stress Test Evaluation for Natural Language Inference. ArXiv. /abs/1806.00692

**Reasons To Accept:**

1. The paper is well-organized. The authors clearly state the problem and their contribution.
2. The proposed method can discover and provide a quantitative analysis to the shortcuts. A natural idea that could benefit future research on shortcut reasoning.

**Reasons To Reject:**

1. This paper raises a broad question: shortcut reasoning. While as mentioned in section 2.1, the proposed method only discovers word(token) triggered shortcuts. Such kind of phenomenon may only contribute to a small proportion of shortcut reasoning (shallow heuristics, artifacts). Also, the task is limited to sequence classification. I think it could be better if these settings are clearly stated at the very beginning of the paper (abstract or introduction).
2. The method description is somehow confusing in detail. Some evidence may not support the claim. I have raised some questions below.
3. The experiment section is lack of details. I think it is necessary to provide more experiment details (settings such as which split does OOD dataset use, results such as overall f1 score $F1(D_{OOD}, f)$, more analysis on unknown shortcut reasoning) in order for the paper to be accepted (it is fine to place them in the appendix). In addition, There is also no mention of the code being released.

**Reproducibility:**

3: Could reproduce the results with some difficulty. The settings of parameters are underspecified or subjectively determined; the training/evaluation data are not widely available.

**Reviewer Confidence:**

3: Pretty sure, but there's a chance I missed something. Although I have a good feel for this area in general, I did not carefully check the paper's details, e.g., the math, experimental design, or novelty.

---

> ### Author Rebuttal · Authors · 2023-08-29
>
> **Thank you for the review and interesting question about the results. We have replied to comments and introduced the result of an additional experiment below.**
>
> ## Reason To Reject
> ### (1): Proposed method only discovers word(token) triggered shortcuts
> As you pointed out, our current method is restricted to discovering token-based shortcuts for classification of a sequence. We will clarify the scope of our method in Introduction. However, we successfully discover many token-based shortcuts without any human intervention, which is a significant achievement.
>
> ### (2): The method description is somehow confusing in detail
> Answered below.
>
> ### (3): The experiment section is lack of details
> We are going to add more details about the experiments in the camera ready version. We will release all the codes and datasets used in the experiments for better reproducibility.
>
> ## Questions For The Authors
> ## Question about method details
> ### (1): word or token
> Our description may be confusing. We use tokens for $w_i$, not words. We will clarify it in the camera-ready version.
>
> ### (2): Masking algorithm
> Yes, IR is to reduce the *less important tokens* first, so that the final shortcut reasoning pattern is shorter. IG scores higher values for a token that strongly influences the prediction. We first mask tokens with lower scores, namely less important tokens for prediction, and then increase masks. We are planning to add an illustration of how IR works in the final version.
>
> ### (3): The last token of IR iteration
> We sequentially mask input sentences to obtain minimal sequences of tokens, starting from the lowest IG scores. Namely, a token with the highest score remains at the end of IR iteration. Since we never know the “expected” prediction after the last token is masked (i.e., a sequence of input-length <mask>), we do not observe whether the prediction flips after the final masking. As such, we leave the last token and output it as a candidate of inference patterns. Importantly, it is a “candidate” of the pattern. We then evaluate the patterns by examining its generality and the extent to which the pattern harms performance on OOD. Therefore, there is no significant concern in utilizing such patterns.
>
> ### (4): what will happen if $\hat{y} \neq \hat{y}′$
> First of all, we want to clarify that $\hat{y}$ is a model’s prediction for original input and $\hat{y}’$ denotes a prediction for a masked input. So, $\hat{y} \neq \hat{y}′$ does not mean wrong prediction in the IID dataset.
> To answer the question, we just ignore the pattern, when the prediction for original input from IID is wrong. We will clarify it in the camera-ready version.
>
> ## Question about soundness
> ### (1):  Is only one example ($l_i=y_i$) enough to say “performing well on IID”?
> Thank you for pointing out this important issue. We admit that $l_i=y_i$ may not be a robust metric for evaluating the efficacy of inference patterns on an IID dataset.
>
> Therefore, we conducted an additional experiment, where we evaluated the performance of each inference pattern using all the instances from IID (as opposed to one example). Specifically:
>
> 1. We define a new metric called iid_acc, which counts the right prediction for inputs that contain trigger (w) as follows:
> $$
> \displaystyle \mathrm{iid} \mathrm{acc} (p_i) \stackrel{\mathrm{def}}{=} \frac{\sum_{x \in E_{iid}(w_i)}\mathbb{1}\left[ f(x)=l_i \land l_i=y_i \right]}{\sum_{x \in E_{iid}(w_i)} \mathbb{1}\left[ f(x)=l_i \right]}.
> $$
> 2. We identify shortcut reasoning from inference patterns based on the original condition for OOD and new metrics for IID. An inference pattern with more than 0.7 of iid_acc is classified as shortcuts. Specifically, we modified Equation (5) (Line 202) as follows:
> $$
> \tilde{P} \stackrel{\mathrm{def}}{=} \\{p_i \in C \mid g(p_i) > \lambda, \mathrm{iid}\mathrm{acc}(p_i) > 0.7, \Delta_i \\! < 0 \\} .
> $$
>
> We observe the results hold a similar trend to the previous results in the paper.  For NLI, negation and some newly discovered shortcuts are observed. For sentiment analysis, sentiment words tend to appear as shortcut reasoning. The results are shown below (Note that if $E$ is more than 1000, we randomly sample 1000 examples from it. So the results are slightly different from the submitted version.):
>
> **NLI**
> | $\tilde{p}$  | $g$ | $\operatorname{iid acc}$ | $\Delta$ | $\mid E \mid$ | $\text{train/test}$ |
> |-----------------|--------------:|---------------:|--------------:|---------------:|:--------------:|
> | [“/s”, "never"] $\rightarrow$ CONTRADICTION | 80.3         | 99.3          | -9.0         | 1515 | x/x |
> | [“/s”, "no"] $\rightarrow$ CONTRADICTION | 69.0         | 94.1          | -6.7         | 2038 | x/x |
> | [“/s", "soon"] $\rightarrow$ NEUTRAL | 64.1         | 100.0         | -6.5         | 142 | x/x |
> | [“/s”, "is", "always"] $\rightarrow$ NEUTRAL | 60.8         | 96.6          | -5.2         | 102 | x/- |
> | [“/s”, "not"] $\rightarrow$ CONTRADICTION | 55.0         | 94.5          | -55.6        | 8708 | x/x |
>
> **SA**
> | $\tilde{p}$  | $g$ | $\operatorname{iid acc}$ | $\Delta$ | $\mid E \mid$ | $\text{train/test}$ |
> |-----------------|--------------:|---------------:|--------------:|---------------:|:--------------:|
> | ["worst"] $\rightarrow$ NEGATIVE | 97.5 | 81.7  | -25.4 | 158 | x/- |
> | ["Excellent"] $\rightarrow$ NEGATIVE | 96.2 |100.0 | -7.2 | 184 | -/x |
> | ["Perfect"] $\rightarrow$ POSITIVE | 96.0 | 88.6 | -12.9 | 324 | -/x |
> | ["poor"] $\rightarrow$ NEGATIVE | 87.1 | 76 | -12.9 | 458 | x/- |
>
>
> We will add the new metrics $\mathrm{iid acc}$, the new Equation (5), and new obtained shortcuts in the camera-ready version.
>
> ### (2): Negation is difficult for NLP models
> We appreciate your interesting analysis. We agree that NLP models tend to struggle to understand negation expressions in general and may not be confident at samples containing negation expressions.
>
> However, we would like to clarify that g actually denotes *the generality of inference patterns (i.e., how much the inference pattern is dominant on the OOD dataset)*, not the performance of a model (cf. Eq.3). In other words, a lower g does not mean that the model is not confident for the input.

---

### Official Review · Reviewer_X4Zv · 2023-08-03

**Typos Grammar Style And Presentation Improvements:** 1. The algorithm description at LINE …
**Soundness:** 3

**Excitement:**

3: Ambivalent: It has merits (e.g., it reports state-of-the-art results, the idea is nice), but there are key weaknesses (e.g., it describes incremental work), and it can significantly benefit from another round of revision. However, I won't object to accepting it if my co-reviewers champion it.

**Paper Topic And Main Contributions:**

This work presents a new method to identify shortcut reasoning patterns in the models. Specifically, the method includes three steps. First, it finds input patterns by using Integrated Gradients to sort the tokens and by gradually masking the input. Then, for each discovered pattern, it calculates the generality of the pattern on the testing set, as well as the performance drop caused by the pattern. In the experiments, the proposed method successfully discovers patterns in both NLI and sentiment analysis datasets.

**Questions For The Authors:**

1. For the train/test column in Table 1, does the train/test refer to the train/test split of the IID dataset? If so, please add clarification in the paper. It can be confusing as the OOD dataset is also another kind of test set.

**Reasons To Accept:**

1. The proposed method is conceptually straightforward and reasonable. It can be easily implemented on many different datasets.
2. On both NLI and sentiment analysis datasets, the method discovers interesting dataset patterns.

**Reasons To Reject:**

1. The pattern discovery in the proposed method heavily relies on the accessibility of an OOD dataset, which is a pretty strong assumption. One main motivation behind all the shortcut discovery work is to figure out and fix the model's weakness so that it can perform well in future unknown OOD settings. Assuming the accessibility of the OOD dataset greatly limits the impact and usefulness of the proposed method. Additionally, this assumption also seems to be one of the key differences between this work and prior work such as Wang et al., 2022a. Since the current draft contains an in-depth comparison of this work and Wang et al., 2022a, I think a discussion of the adoption and the related consequences of this assumption (instead of just one sentence in the limitations section) is needed for this paper.
2. Due to the mask searching step, the method is computationally expensive (as acknowledged in LINE 235-236).

**Reproducibility:**

4: Could mostly reproduce the results, but there may be some variation because of sample variance or minor variations in their interpretation of the protocol or method.

**Reviewer Confidence:**

4: Quite sure. I tried to check the important points carefully. It's unlikely, though conceivable, that I missed something that should affect my ratings.

---

> ### Author Rebuttal · Authors · 2023-08-29
>
> **Thank you for the review and advice for the paper. We have organized our responses below.**
>
> ## Reason To Reject
> ### (1-1): Proposed method heavily relies on the accessibility of an OOD dataset
>
> When an OOD dataset is not available, one could use LLMs to generate OOD data. Now that we can access language models that have surprising linguistic capabilities and are well-aligned with the user's instruction. The generated examples by LLMs have a certain distribution which can be treated as OOD for target models, or we can prompt them to generate examples with specific distribution.
>
> As described below, even the SoTA method still relies on OOD dataset, and discovering shortcuts without the accessibility of OOD dataset has considerable difficulty. We think that this exploration is beyond the scope of this paper.
>
> While we already acknowledge the difficulty about the accessibility of OOD datasets in Limitation section, we will enrich the Limitation section by adding more discussion like this in the final version.
>
> ### (1-2): Prior work such as Wang et al. 2022a didn’t rely on an OOD dataset
> We would like to point out that Wang et al. (2022a) also use an OOD dataset. They propose cross-dataset analysis based on two types of datasets:(i) the one used for training a model and (ii) the other one from different sources or domains used for discovering shortcut reasoning with the first dataset (called “cross-dataset analysis”). The second dataset corresponds to our OOD dataset. We think that Wang's method and our method are not different on this point, that is, both methods require two datasets.
>
> ### (2): Proposed method is computationally expensive
> The main use case of our method is as follows. During the development cycle, developers create an NLP model, and they apply our method to the model to find the model’s weakness. This is done only during the development phase, and they don’t need to use our method after deployment. As you pointed out, our method does need extensive computation, but it is not prohibitively costly. For example, on 1000 examples of sentiment analysis data, Input Reduction took only 2 min 29 sec, and Generality calculation, which is the most computationally expensive, took only 47 min 27 sec on NVIDIA A6000 48GB.
>
> We suspect that the term “Computational limitation” in Line 235-236 was misleading. This was supposed to mean specifically about run time. Unless the user cares about time, our framework can handle more than 1000 examples.
>
> We clarify all the above points in the final version.
>
> ## Questions For The Authors
> ### (1): Train/test column
> You are right. We are going to add some explanation on it.
>
> ## Typos Grammar Style And Presentation Improvements
> ### (1): Description of IR is confusing
> Thank you for the suggestion. We are planning to clarify this  by adding a figure showing a working example (e.g., “Spielberg is a great director” =>
> “Spielberg is <MASK> great director” => “Spielberg <MASK> <MASK> great director” => …) in the camera-ready version.

---

### Official Review · Reviewer_iQTs · 2023-08-03

**Typos Grammar Style And Presentation Improvements:** 1. Line 333, "unless" should be "if".…
**Soundness:** 3

**Excitement:**

3: Ambivalent: It has merits (e.g., it reports state-of-the-art results, the idea is nice), but there are key weaknesses (e.g., it describes incremental work), and it can significantly benefit from another round of revision. However, I won't object to accepting it if my co-reviewers champion it.

**Paper Topic And Main Contributions:**

This paper focuses on the task of discovering shortcut reasoning from an NLP model. Shortcut reasoning is the "lazy thinking" of a NLP model, in other words, the irrational process of inference, which affects the robustness of an NLP model. This irrational process of inference is often derived from the spurious correlations in the training data.  For instance, for sentiment analysis, a model tends to classify a movie review with word "Spielberg" is positive. However, "Spielberg" should not be a causal feature for the classification.

The contribution of the paper are:
1. Different from most of the previous work (except for (Wang et al. 2022a)), which based on (1) human priors and (2) error analysis over the model (3) a pre-defined series of test types,  this paper proposed an automatic method for identifying shortcut reasoning.

2. This method can quantify the severity of shortcut reasoning by leveraging OOD data.
3. This automatic process can not only discover reported ones in previous research but also can discover unknown shortcut reasoning.

**Questions For The Authors:**

Question A:
In Appendix A line 11, since the while conditon is $\hat{y}$ == $\hat{y} ' $, then after the while condition $\hat{y} ' $ has already flipped, so line 11 actually include ["Spielberg"] → negative into the shortcut reasoning set, rather than ["Speilberg"] → positive, which should be included.
Is my understanding correct? It looks like an issue needs to be changed.

Question B:
Line 343-345, what does the following statement mean?
> "If the prediction for the masked input does not flip during the reduction, then we alternatively output the last token left in the input".

Why it is processed in this way?  You can do in the way that not include this shortcut reasoning right? Also this process is not reflected in Appendix A, which is confusing.

Question C:
In Line 152 - 155, "IR sorts the tokens …. ascending order of their rank". This statement is not clear. How the tokens $x_i$ is sorted based on IG score, ascending or descending? Also gives a brief explanation of IG score, the higher the more important or the other way? So the IR is to reduce the less important tokens first, so that the final shortcut reasoning pattern is shorter?

Question D:
Will the sampled test data be released for reproduce the result?

**Reasons To Accept:**

This paper is the second paper that can automatically discover shortcut reasoning. It addressed the following limitation of the first paper (Wang et al, 2022a):

(1) Wang et al, 2022a lacks the method to quantify the severity of the discovered shortcut reasoning on OOD data. This method is necessary and interesting, since we are concerned with how the shortcut reasoning will affect the potential OOD data.

(2) (Wang et al, 2022a) made a plausible assumption that _genuine_ tokens which are useful for predicting labels across different datasets do not lead to shortcut reasoning. However, (Joshi et al. 2022) argue that this assumption does not hold in the situation that these tokens alone may not provide sufficient information to accurately predict labels, for instance, in the sentence "This movie is not good", "good" is necessary but not sufficient to determine the sentiment of the sentence.

(3) Since this paper can quantify the severity of the shortcut, it can rank and show the representative shortcut reasoning. In the experiment part, it shows the most representative shortcut reasoning and verifies that the result aligns with the findings mentioned in previous works (shown in Sec 3.2 when discussing dataset NLI and SA).

(4) Besides, the presentation and writing of this paper is very concise and clear, easy to follow and understand.

**Reasons To Reject:**

(1)  The main weakness of this paper is the lack of trustworthy evaluation involving human annotation.

Evaluating identified shortcuts in machine learning or deep learning based on models is difficult. The work Wang, et al, 2022a gives very detailed and convincing evaluation involving human annotators and provides precision scores. Although this work identifies some shortcut reasoning that aligns with the findings in the previous research, it lacks an overall and trustworthy evaluation on all the shortcut reasoning it has discovered. I hope it can also show more case studies, demonstrations of the examples.

I hope the evaluation issue can be addressed so that I would like it to be shown in the main conference short paper. Otherwise, I feel that the evaluation is not sufficient.

**Reproducibility:**

4: Could mostly reproduce the results, but there may be some variation because of sample variance or minor variations in their interpretation of the protocol or method.

**Reviewer Confidence:**

5: Positive that my evaluation is correct. I read the paper very carefully and I am very familiar with related work.

---

> ### Author Rebuttal · Authors · 2023-08-29
>
> **Thank you for the detailed review. We respond to your comments and questions below.**
>
> ## Reason To Reject
> ### (1): The lack of trustworthy evaluation involving human annotation
> First of all, we would like to clarify the difference between Wang et al. (2022a)’s goal and ours. Wang et al.(2022a) aimed to identify shortcut reasoning that is irrational from humans’ point of view. Thus, they conducted human evaluation for the verification of the results.
>
> On the other hand, our work employs less subjective definition: we focus on detecting shortcut reasoning that performs well on IID and underperforms on OOD (Lines 181-185; see Geirhos et al. (2020) for the detailed definition and discussion). Hence, we do not care whether the detected shortcut reasoning is irrational or not from humans’ point of view. In our work, we construct quantitative metrics (e.g., $g$, $\Delta$) for each definition above, where they do not require any human annotators.
>
> Overall, we believe that the discovered shortcut reasoning is already trustworthy because they strictly align with the definitions. We will clarify these points in the final version.
>
> ## Questions For The Authors
> ### (A): An error in pseudo-code in Algorithm 1 in Appendix A
> Thank you for pointing out the fault of the pseudo-code. We found that the pseudo-code was indeed incorrect and needed the following revision:
> - line 6: $\hat{y}'\_{prev} \gets f(x'_{prev}) \\, ; \\, \hat{y}' \gets f(x')$
> - line 11: $C \gets C \cup \\{ p = (x'\_{prev},\hat{y}'\_{prev}) \\}$
>
> Therefore, Algorithm 1 does add inference patterns with an unflipped label to C (e.g., [“Spielberg”] -> positive, in your example). We will clarify it in the final version. Importantly, we note that only the pseudo-code was incorrect and the reported results had no problem.
>
> ### (B): The last token of IR iteration
> We sequentially mask input sentences to obtain minimal sequences of tokens, starting from the lowest IG scores. Namely, a token with the highest score remains at the end of IR iteration. Since we never know the “expected” prediction after the last token is masked (i.e., a sequence of input-length <mask>), we do not observe whether the prediction flips after the final masking. As such, we leave the last token and output it as a candidate of inference patterns. Importantly, it is a “candidate” of the pattern. We then evaluate the patterns by examining its generality and the extent to which the pattern harms performance on OOD. Therefore, there is no significant concern in utilizing such patterns.
>
> ### (C): The definition of IG score
> We agree that the description (Line 152-155) is unclear. As you pointed out, it is correct that the IR is to reduce the *less important tokens* first, so that the final shortcut reasoning pattern is shorter. IG scores higher values for a token that strongly influences the prediction. We first mask tokens with lower scores, namely less important tokens for prediction, and then increase masks. We are planning to add an illustration of how IR works in the final version.
>
> ### (D): Sampled data
> Yes, we will release all the codes and datasets used in the experiments, including the sampled data, for better reproducibility.
>
> ## Typos Grammar Style And Presentation Improvements
> ### (1): “Unless” is wrong?
> Yes, “if” is right.

---

### Meta-Review · Area_Chair_vTes · 2023-09-12

**Recommendation:** 3

**Metareview:**

TLDR: The paper makes an impportant contribution proposing an automatic method for identifying shortcut reasoning; it can quantify the severity of the shortcut, and discover unknown shortcut reasoning. The concerns raised by the reviewrs are mainly about limitations of the work and a better comparison with related work. There are also some clarification points. The authors address well all raised concerns in their rebuttal and provide additional results to further support their claims.

Following is a summary of the pros and cons identified by the reviewers in order of importance:

Pros:
1. Significance/Originality - all reviewers agree that the paper makes an important contribution with an automatic method for identifying shortcut reasoning; it addresses important limitations of related work on shortcut discovery (iQTs), the method discovers interesting dataset patterns (X4Zv), could benefit future research on shortcut reasoning (xMKL).
2. Quality - the proposed method is conceptually straightforward and reasonable (X4Zv).
3. Clarity - the presentation and writing of this paper are very concise and clear (iQTs, xMKL).

Cons:
1. Clarity - some evidence might not support the claim, e.g., is only one example enough to say performing well on IID (xMKL).
2. Clarity - include a more comprehensive discussion of the limitations of the method - the need for OOD dataset (X4Zv), computational cost (X4Zv), discovering only a certain type of shortcut reasoning (xMKL).
3. CLarity - a more comprehensive comparison with related work in terms of evaluation (iQTs), and dataset usage (X4Zv).

---

### Decision · Program_Chairs · 2023-10-07

**Decision:**

Accept-Findings

**Comment:**

TLDR: The paper makes an impportant contribution proposing an automatic method for identifying shortcut reasoning; it can quantify the severity of the shortcut, and discover unknown shortcut reasoning. The concerns raised by the reviewrs are mainly about limitations of the work and a better comparison with related work. There are also some clarification points. The authors address well all raised concerns in their rebuttal and provide additional results to further support their claims.

Following is a summary of the pros and cons identified by the reviewers in order of importance:

Pros:
1. Significance/Originality - all reviewers agree that the paper makes an important contribution with an automatic method for identifying shortcut reasoning; it addresses important limitations of related work on shortcut discovery (iQTs), the method discovers interesting dataset patterns (X4Zv), could benefit future research on shortcut reasoning (xMKL).
2. Quality - the proposed method is conceptually straightforward and reasonable (X4Zv).
3. Clarity - the presentation and writing of this paper are very concise and clear (iQTs, xMKL).

Cons:
1. Clarity - some evidence might not support the claim, e.g., is only one example enough to say performing well on IID (xMKL).
2. Clarity - include a more comprehensive discussion of the limitations of the method - the need for OOD dataset (X4Zv), computational cost (X4Zv), discovering only a certain type of shortcut reasoning (xMKL).
3. CLarity - a more comprehensive comparison with related work in terms of evaluation (iQTs), and dataset usage (X4Zv).